

# 24-hour cloud cover calculation using ground-based imager with machine learning

Bu-Yo Kim, Joo Wan Cha, Ki-Ho Chang

Convergence Meteorological Research Department, National Institute of Meteorological Sciences, Seogwipo, Jeju 63569, Republic of Korea

*Correspondence to*: Bu-Yo Kim (kimbuyo@korea.kr)

**Abstract.** In this study, image data features and machine learning methods were used to calculate 24-h continuous cloud cover from image data obtained by a camera-based imager on the ground. The image data features were the time (Julian day and hour), solar zenith angle, and statistical characteristics of the red-blue ratio, blue–red difference, and luminance. These features were determined from the red, green, and blue brightness of images subjected to a pre-processing process involving masking removal and distortion correction. The collected image data were divided into training, validation, and test sets and were used to optimize and evaluate the accuracy of each machine learning method. The cloud cover calculated by each machine learning method was verified with human-eye observation data from a manned observatory. Supervised machine learning models suitable for nowcasting, namely, support vector regression, random forest, gradient boosting machine, k-nearest neighbor, artificial neural network, and multiple linear regression methods, were employed and their results were compared. The best learning results were obtained by the support vector regression model, which had an accuracy, recall, and precision of 0.94, 0.70, and 0.76, respectively. Further, bias, root mean square error, and correlation coefficient values of 0.04 tenth, 1.45 tenths, and 0.93, respectively, were obtained for the cloud cover calculated using the test set. When the difference between the calculated and observed cloud cover was allowed to range between 0, 1, and 2 tenths, high agreement of approximately 42%, 79%, and 91%, respectively, were obtained. The proposed system involving a ground-based imager and machine learning methods is expected to be suitable for application as an automated system to replace human-eye observations.

## 1 Introduction

To date, ground-based cloud cover observation has been performed using the human eye, in accordance with the normalized synoptic observation rule of the World Meteorological Organization (WMO), and recorded in tenths or oktas (Kim et al., 2016; Yun and Whang, 2018). However, human-eye observation of cloud cover lacks consistency and depends on the observer conditions and the observation term (Mantelli Neto et al., 2010; Yang et al., 2016). Further, although continuous cloud cover observation during both day and night is important, current data continuity is lacking because longer-period observations are



performed at night rather than during the day (Kim et al., 2020b). In addition, construction of a dense cloud observation network from observation environments with low accessibility, such as mountaintops, is difficult. Therefore, meteorological satellites and ground-based remote observation equipment that can continuously monitor clouds while overcoming these problems are now being employed (Yabuki et al., 2014; Yang et al., 2015; Kim et al., 2016, 2020b).

Geostationary satellites can observe clouds on the global scale at intervals of several minutes; however, their spatial resolution
is as large as several kilometers (Kim et al., 2018b; Lee et al., 2018). Polar satellites have spatial resolutions of several hundred meters, i.e., high resolution; however, they can observe the same area only once or twice per day (Kim et al., 2019, 2020a). For both geostationary and polar satellites, geometric distortion problems occur during cloud cover estimation on the ground, depending on the cloud height (Mantelli Neto et al., 2010). As cloud heights and thicknesses vary, the cloud detection uncertainty also varies depending on the position of the sun or satellite (Ghonima et al., 2012). In general, cloud cover
estimation using satellite data differs from the approach used for human-eye observation data, because the wide grid data around the central grid are averaged or calculated as fractions (Alonso-Montesinos, 2020; Sunila et al., 2021).

Radar, LiDAR, ceilometers, and camera-based imagers can be used as ground-based observation instruments (Long et al., 2006; Shields et al., 2013). With regard to radar, cloud radar technology such as Ka-band radar is suitable for cloud detection but has the disadvantage of reduced detection accuracy with increased distance from the radar apparatus (Kim et al., 2020c;
Yoshida et al., 2021). For LiDAR and ceilometers, the uncertainty is very large because the cloud cover is calculated from the signal intensity of a narrow portion of the sky (Peng et al., 2015; Kim et al., 2020b). In contrast, for a camera-based imager, the sky in the surrounding hemisphere can be observed through a fisheye lens (180° field of view (FOV)) mounted on the camera. Further, depending on the performance of the imager and the operation method, clouds can be observed continuously for 24-h, i.e., through the day and night. The data can be stored as images and the cloud cover can be calculated from these
data (Kim et al., 2020b; Sunila et al., 2021).

Many studies have attempted to use camera-based imagers for automatic cloud observation and cloud cover calculation on the ground (Dev et al., 2016; Lothon et al., 2019; Shields et al., 2019). Those results can be used for numerical weather analysis and forecasting; they are also very economical and ideal for cloud monitoring over local areas (Mantelli Neto et al., 2010; Kazantzidis et al., 2012; Ye et al., 2017; Valentín et al., 2019). In general, cloud cover can be calculated based on the brightness
of the red, green, and blue (RGB) colors of the image taken by the imager. In detail, the RGB brightness varies according to the light scattering from the sky and clouds and, using the ratio or difference between these colors, cloud can be detected and cloud cover can be calculated (Liu et al., 2015; Yang et al., 2015; Kim et al., 2016). For example, when the red-blue ratio (RBR) is 0.6 or more or the red–blue difference (RBD) is less than 30, the corresponding cloud cover is classified (i.e., using a threshold method) as a cloud pixel and incorporated in the cloud cover calculation (Kruter et al., 2009; Heinle et al., 2010; Liu
et al., 2015; Azhar et al., 2021). However, using these empirical methods, it is difficult to distinguish between the sky and clouds under various weather conditions (Yang et al., 2015). This is because the colors of the sky and clouds vary with the atmospheric conditions and because the sun position and threshold conditions can change continuously (Yabuki et al., 2014; Blazek and Pata, 2015; Cazorla et al., 2015). Therefore, methods of cloud detection and cloud cover calculation involving



application of machine learning methods to images are now being implemented, as an alternative to empirical methods (Peng et al., 2015; Lothon et al., 2019; Al-lahham et al., 2020; Shi et al., 2021).

Cloud cover can be calculated from camera-based imager data using a supervised machine learning method capable of regression analysis (Al-lahham et al., 2020). Supervised learning is a method through which a prediction model is constructed using training data which already contain the labeled data. Examples include support vector machines (SVMs), decision trees (DTs), gradient boosting machines (GBMs), and artificial neural networks (ANNs) (Çınar et al., 2020; Shin et al., 2020). Deep learning methods that repeatedly learn data features by sub-sampling image data at each convolution step for gradient descent are also available, such as convolutional neural networks (Dev et al., 2019; Shi et al., 2019; Xie et al., 2020). However, this approach is difficult to utilize for nowcasting because considerable physical resources and time are consumed by the learning and prediction processes (Al Banna et al., 2020; Kim et al., 2021).

In this study, 24-h continuous cloud cover is calculated from image data obtained by a camera-based imager on the ground, using image data features and machine learning methods. ANN, GBM, $k$-nearest neighbor (kNN), multiple linear regression (MLR), support vector regression (SVR), and random forest (RF) methods suitable for nowcasting are used to calculate cloud cover continuously for 24 h using data from the ground-based imager. For each of these methods, an optimal prediction model is constructed by setting hyper-parameters. The machine learning model most suitable for cloud cover calculation is then selected by comparing the prediction performance of each model on training and validation datasets. The cloud cover calculated from the selected machine learning model is then compared with human-eye observation data and the results are analyzed. The remainder of this paper is organized as follows. The image and observation data used in this study are described in Sect. 2, and the machine learning methods and their sets are summarized in Sect. 3. The prediction performance evaluation for each machine learning method and the calculation result verification are reported in Sect. 4. Finally, the summary and conclusion are given in Sect. 5.

## 2. Research data and methods

### 2.1 Ground-based imager

In this study, a digital camera-based automatic cloud observation system (ACOS) was developed using a Canon EOS 6D camera to detect and calculate cloud cover for 24-h, as shown in Fig. 1. This system was developed by the National Institute of Meteorological Sciences (NIMS)/Korea Meteorological Administration (KMA) and A&D·3D Co., Ltd. (Kim et al., 2020b). The ACOS was installed at the Daejeon Regional Office of Meteorology (DROM; 36.37°N, 127.37°E), a manned observatory in which cloud cover observation by human eye is performed. The detailed ACOS specifications are listed in Table 1. The International Organization for Standardization (ISO) values of the complementary metal oxide semiconductor (CMOS) sensor employed in the camera are 100 (day)–25600 (night), and the sensitivity is adjusted according to the image brightness. In this study, the camera shutter speed was set to 1/1,000 s (day)–5 s (night), considering the long exposure for object detection required at night. The F-stop was set to F8 (day)–F11 (night), and the sky-dome object was taken with a large depth of field



(Peng et al., 2015; Dev et al., 2017). The camera lens was installed at a height of 1.8 m, similar to human-eye height, and a fisheye lens (EF8-15 F/4L fisheyes USM) was installed to capture the entire surroundings, including the sky and clouds, within a 180° FOV. To perform 24-h continuous observation, heating (below –2 °C) and ventilation devices were installed inside the ACOS body to facilitate image acquisition without manned management (Dev et al., 2015; Kim et al., 2020b).


a)                                                        b)

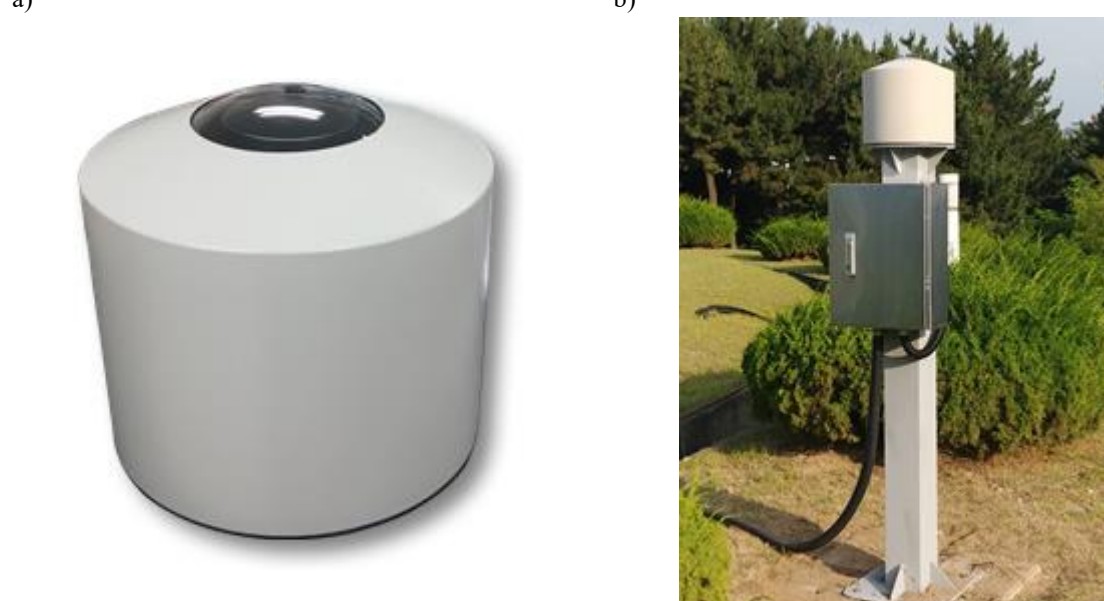

**Figure 1: ACOS appearance (a) and installation environment (b) (Kim et al., 2020b).**

**Table 1: Detailed ACOS specifications.**

| Function | Description |
| --- | --- |
| Size | 264 mm (L) x 264 mm (W) x 250 mm (H), 6.5 kg |
| Pixels | 2,432 x 2,432 |
| Focal length | 8 mm, 180° fisheye lens |
| Sensor | CMOS |
| Aperture | F8 (day)–F11 (night) |
| Sutter speeds | 1/1,000s (day)–5s (night) |
| ISO | 100 (day)–25600 (night) |
| Observation periods | 24-h operation, 10-min interval observation |
| Etc. | 24-h automatic heating (below –2°C) and ventilation |




## 2.2 Cloud cover calculation and validation

The image data captured by ACOS were processed by converting each RGB channel of each image pixel to a brightness of 0–255. Although the camera-lens FOV was 180°, only pixel data within the zenith angle of 80° (FOV 160°) were used. This condition was in consideration of the permanent masking area of the horizontal plane due to surrounding objects (buildings,
trees, equipment, etc.) (Kazantzidis et al., 2012; Shields et al., 2019; Kim et al., 2020b). For cloud cover calculation using the ACOS images, image data taken at 1-h intervals from January 1 to December 31, 2019, were used. The cloud cover was calculated using the statistical characteristics of the RGB brightness ratio (i.e., the red-blue ratio (RBR)), difference (i.e., the blue–red difference (BRD)), and luminance (Y), which vary for each image (Sect. 2.3), as well as supervised machine learning methods (Sect. 3). The calculated cloud cover was compared with human-eye observation data from DROM. As the cloud
cover was calculated as a percentage between 0 and 100%, the result was converted to an integer (tenth) between 0 and 10 (Table 2) for comparison with the human-eye-based cloud cover values. As the ACOS was installed at DROM, there were no location differences between observers; thus, the same clouds were captured (Kim et al., 2020b). At DROM, night observations were performed at 1-h intervals during inclement weather (rainfall, snowfall, etc.), but otherwise at 3-h intervals. The night period varied with the season. Considering this, a total of 7,402 images were collected, excluding missing cases, from the
ACOS.

**Table 2: ACOS cloud cover (%) to DROM human-eye-observed cloud cover (tenths) conversion table.**

| % | ≤ 5 | 5~15 | 15~25 | 25~35 | 35~45 | 45~55 | 55~65 | 65~75 | 75~85 | 85~95 | 95 < |
|---|---|---|---|---|---|---|---|---|---|---|---|
| Tenth | 0 | 1 | 2 | 3 | 4 | 5 | 6 | 7 | 8 | 9 | 10 |

The entire collected dataset was randomly sampled without replacement. Overall, 50% (3,701 cases) of the total data elements were configured as a training set, 30% (2,221 cases) as a validation set, and 20% (1,480 cases) as a test set (Xiong et al., 2020). The training set was used to train the machine learning algorithms, and the prediction performance of each machine learning method was assessed using the validation set. Optimal hyper-parameters were set for each machine learning method through the training and validation sets. The results of each machine learning method were compared. In this process, the test set was
input to the machine learning model that exhibited the best prediction performance, and the calculated results and human-eye observation data were compared. The accuracy, recall, precision, bias, root mean square error (RMSE), and correlation coefficient (R) were analyzed according to Eqs. (1)–(6); hence, the prediction performance of each machine learning method was determined and compared based on the human-eye observation data.





$$Accuracy = \frac{TP + TN}{TP + TN + FP + FN} \tag{1}$$

$$Recall = \frac{TP}{TP + FN} \tag{2}$$

$$Precision = \frac{TP}{TP + FP} \tag{3}$$

$$Bias = \frac{\sum(M - O)}{N} \tag{4}$$

$$RMSE = \sqrt{\frac{\sum(M - O)^2}{N}} \tag{5}$$

$$R = \frac{\sum(M - \overline{M})(O - \overline{O})}{\sqrt{\sum(M - \overline{M})^2}\sqrt{\sum(O - \overline{O})^2}} \tag{6}$$


Here, *TP*, *TN*, *FP*, and *FN* are the number of true positives (reference: yes, prediction: yes), true negatives (reference: yes, prediction: no), false positives (reference: no, prediction: yes), and false negatives (reference: no, prediction: no), respectively. Further, *M*, *O*, and *N* are the cloud cover calculated by the employed machine learning method, the human-eye-observed cloud cover, and the number of data, respectively.


## 2.3 Machine learning input data

The data input to the machine learning algorithms for cloud cover calculation using the ACOS images were produced as follows. First, as the ACOS image was taken with a fisheye lens, the image was distorted. That is, objects at the edge were smaller than those at the center of the image (Chauvin et al., 2015; Yang et al., 2015; Lothon et al., 2019). Therefore, the relative size of

each object in the image was adjusted through orthogonal projection distortion correction according to the method expressed in Eqs. (7)–(11) (Kim et al., 2020b). Second, surrounding masks such as buildings, trees, and equipment, as well as light sources such as the sun, moon, and stars, were removed from the image (masking was performed when the mean RGB brightness exceeded 240). These objects directly mask the sky and clouds or make it difficult to distinguish them; therefore, they must be removed when calculating cloud cover (Yabuki et al., 2014; Kim et al., 2016, 2020b). Third, the RBR, BRD, and

Y frequency distributions were calculated using the RGB brightness of each pixel of image data subjected to pre-processing (i.e., masking removal and distortion correction). Here, Y was calculated as Y = 0.2126R+0.7152G+0.0722B (Sazzad et al., 2013; Shimoji et al., 2016). The class interval sizes of the RBR, BRD, and Y frequency distributions were set to 0.02, 2, and





2, respectively, and classes with frequencies of less than 100 were ignored. The mean, mode, frequency of mode, kurtosis, skewness, and quantile (Q0–Q4: 0%, 25%, 50%, 75%, and 100%) data were obtained for each frequency distribution, and

these data, the time information (Julian day and hour), and solar zenith angle (SZA) data were used as input data for each machine learning method. Note that consideration of the Julian day and hour allowed distinction between the season and day and night. Further, the SZA should be considered because the colors of the sky and clouds change according to the position of the sun (Blazek and Pata, 2015; Cazorla et al., 2015; Azhar et al., 2021). As these image data features have different appearances under different conditions (cloud cover, day, night, etc.), they constitute an important variable in machine learning

regression for cloud cover calculation (Heinle et al., 2010; Li et al., 2011).

$$r = \sqrt{(x - cx)^2 + (y - cy)^2} \qquad (7)$$

$$\theta = \mathrm{asin}\,(r/radi) \qquad (8)$$

$$\phi = \mathrm{asin}\,((y - cy)/r) \qquad (9)$$

$$x' = cx + r \times \theta \times \cos(\phi) \qquad (10)$$

$$y' = cy + r \times \theta \times \sin(\phi) \qquad (11)$$

where $r$ is the distance between the center pixel ($cx$, $cy$) of the original image and each pixel ($x$, $y$), $\theta$ is the SZA, $radi$ is the image radius, $\phi$ is the azimuth, and $x'$ and $y'$ are the coordinates of each pixel after distortion correction.


Figure 2 shows distortion corrected images for day and night cloud-free, overcast, and partly cloudy cases and the RBR, BRD, and Y relative frequency distributions. The frequency distributions were expressed as percentages over approximately 310,000 pixels excluding the masked area. Human-eye observations at DROM yielded cloud-free, overcast, and partly cloudy case values of 0, 10, and 5 tenths, respectively. As for the RBR frequency distribution during the day, larger RBR distributions were

observed for the overcast than cloud-free case, and bimodal distributions including both (i.e., overcast and cloud-free) distributions were obtained for the partly cloudy case. The variance was large in the partly cloudy case. With regard to the BRD frequency distribution, the blue-channel brightness increased with Rayleigh scattering, such that the cloud-free case with many sky pixels had larger BRD distribution than the overcast case (Ghonima et al., 2012; Kim et al., 2016). In contrast, the Y frequency distribution was relatively large for the overcast case, which involved many cloud pixels. Although the RBR

frequency distributions at night and day were similar, the RBR was larger at night because the red-channel brightness increased under the influence of Mie scattering (Kyba et al., 2012; Kim et al., 2020b). A negative BRD distribution was obtained from the cloud pixels. At night, there is no light source such as the sun. Therefore, in this study, RGB brightness close to black (0, 0, 0) was distributed in the cloud-free case, yielding small Y. As the images obtained through ACOS had different RBR, BRD, and Y frequency distribution classes and shapes for each case, it was necessary to train these data features (i.e., the mean, mode,


frequency of mode, kurtosis, skewness, and quantile of each frequency distribution) on a machine learning model to calculate

the cloud cover.



**Figure 2: Distortion corrected images and RBR (d, j), BRD (e, k), and Y (f, l) relative frequency distributions for cloud-free (0 tenth), overcast (10 tenths), and partly cloudy (5 tenths) cases at day and night. The daytime cloud-free (a), overcast (b), and partly cloudy**



**(c) data were obtained at 1400 LST on 8 March, 1200 LST on 15 July, and 1500 LST on 28 September 2019. Cloud-free (g), overcast (h), and partly cloudy (i) nighttime data were obtained at 0300 LST on 24 January, 2000 LST on 18 February, and 2200 LST on 30 April 2019. The green and red areas are masked to remove surrounding masks (i.e., buildings, trees, and equipment) and light sources (i.e., the sun and moon), respectively.**

## 3. Machine learning methods

Depending on the machine learning method, even if the accuracy, recall, precision, and R of the trained model are high and the bias and RMSE are small, overfitting problems may occur when data other than training data are used for prediction; these problems can yield low prediction performance (Ying, 2019). Therefore, in this study, optimal hyper-parameters were set by iteratively changing the hyper-parameter for each machine learning method using the training and validation sets (Bergstra

and Bengio, 2012). The optimal hyper-parameter was determined based on the accuracy, recall, precision, bias, RMSE, and R, which were prediction performance indicators for each iteration. The details and hyper-parameter settings of each supervised machine learning method used in this study are described in Sects. 3.1 to 3.6. The prediction results of each machine learning method are compared in Sect. 4.1.

### 3.1 Support vector regression (SVR)

SVR is an extended method that can be used for regression analysis by introducing an ε-insensitive loss function to an SVM. As shown in Fig. 3a, a hyperplane consisting of support vectors that can classify the maximum margin for the distance between vectors is found (Gani et al., 2010; Taghizadeh-Mehrjardi et al., 2017). The optimal hyperplane is obtained by finding $w$ and $b$ that minimize the mapping function ($\Phi(w)$), as shown in Eq. (12) (Meyer and Wien, 2021). The constraints are shown in Eq.

(13). Then, as in Eq. (14), the kernel is applied and mapped to a higher dimension. Here, $\varepsilon$ determines the threshold of margin, $\xi$ is a slack variable to allow error, and $C$ is the allowable cost that can violate the constraint of Eq. (13). In this study, the R "e1071" package (Meyer et al., 2021) used, the SVR kernel was set as a radial basis function (RBF), and the hyper-parameters were set to *epsilon* ($\varepsilon$) = 0.12, *gamma* ($\gamma$) = 0.04, and *cost* ($C$) = 5.

$$\Phi(w) = \min \frac{1}{2} \|w\|^2 + C \sum_{i=1}^{n} (\xi_i + \xi^*_i) \tag{12}$$

$$(w^T x_i + b) - y_i \leq \varepsilon + \xi_i, y_i - (w^T x_i + b) - y_i \leq \varepsilon + \xi^*_i, \qquad \xi_i, \xi^*_i \geq 0 \tag{13}$$

$$K(x_i, x_j) = \exp\left(-\gamma (x_i - x_j)^2\right) \tag{14}$$




where $x_i$ and $x_j$ are each data point and $\gamma$ is a parameter that controls the RBF kernel width.

### 3.2 Random forest (RF)

The RF method composes $N$ decision trees by combining randomly selected variables from each node to grow a regression
tree, as shown in Fig. 3b. An ensemble of the results of each decision tree is obtained, and hence, a prediction result is provided
(Wright et al., 2017). That is, in the RF ensemble learning method, every individual tree of the decision tree contributes to the
final prediction (Shin et al., 2020; Kim et al., 2021). In this study, the R "Ranger" package (Wright et al., 2018) was used, and
the hyper-parameters were set to *ntree* (the number of trees) = 510, *mtry* (the number of variables randomly sampled from each
node) = 7, *min.node.size* (minimal node size) = 5.

### 3.3 Gradient boosting machine (GBM)

The GBM uses boosting instead of bagging during resampling and ensemble processes. As shown in Fig. 3c, a model with
improved predictive power is created by gradually improving upon the parts that the previous model could not predict while
sequentially generating weak models. The final prediction is calculated from the weighted mean of these results (Friedman,
2001). In other words, gradient boosting updates the weights iteratively to minimize the difference from the function $f(x)$ that
predicts the actual observation using gradient descent (Ridgeway, 2020). In this study, the R "gbm" package (Greenwell et al.,
2020) was used; the GBM kernel was set to a Gaussian distribution function; and the hyper-parameters were set to *n.trees*
(number of trees) = 500, *interaction.depth* (maximum depth of binary tree) = 5, shrinkage (learning rate) = 0.1.

### 3.4 k-Nearest neighbor (kNN)

The kNN method involves non-parametric, instance-based learning, and is one of the simplest predictive models in machine
learning. The kNN algorithm finds the $k$-nearest neighbors to the query in the data feature space, as shown in Fig. 3d, and then
predicts the query with distance-based weights (Zhang et al., 2018b). That is, a set of independent variables is constructed as
a cluster, and values corresponding to each neighbor are weighted according to the Euclidean distance and predicted (Martínez
et al., 2019). In this study, the R "class" package (Ripley and Venables, 2021a) was used and the hyper-parameter setting was
$k = 15$.

### 3.5 Artificial neural network (ANN)

An ANN is a mathematical model that mimics a neuron; i.e., the signal transmission system of a biological neural network. As
shown in Fig. 3e, this model consists of an input layer that receives input data, an output layer that outputs prediction results,



and an invisible hidden layer between the two layers (Rosa et al., 2020). The hidden node of the hidden layer acts like a neuron in a neural network and is composed of weight, bias, and an activation function. In this study, we used the R "nnet" package (Ripley and Venables, 2021b), which is based on feed-forward neural networks with a single hidden layer that can rapidly learn and predict while considering nowcasting. The hyper-parameters of this package were set as follows: *size* (number of hidden

nodes) = 7, *maxit* (maximum number of iterations) = 700, and *decay* (weight decay parameter) = 0.05.

**3.6 Multiple linear regression (MLR)**

The method in which the relationship of the dependent variable to the independent variable is regressed by considering one independent variable only is called simple linear regression, and the method in which the change in the dependent variable is

predicted based on the changes in two or more independent variables is called MLR. An MLR model with $k$ independent variables predicts the dependent variable as shown in Eq. (15), using the least squares method which minimizes the predictor variable and the sum of squared errors (Fig. 3f) (Olive, 2017). In this study, we used the R "glm" package (Geyer, 2003).

$$Y_i = \beta_0 + \beta_1 X_{1i} + \beta_2 X_{2i} + \cdots + \beta_k X_{ki}, \qquad i = 1, 2, \cdots, N \tag{15}$$

where $\beta_k$ are the population coefficients (i.e., parameters), and $X_{ki}$ is the $k$-th predictor of the $i$-th observation (a value that describes the variable $Y_i$ to be predicted). In this study, the independent variables were the RGB, BRD, and Y mean, mode, frequency of mode, skewness, kurtosis, quantile, as well as the Julian day, hour, and SZA. The dependent variable was the cloud cover observed by human eyes.





a) Support Vector Regression (SVR)

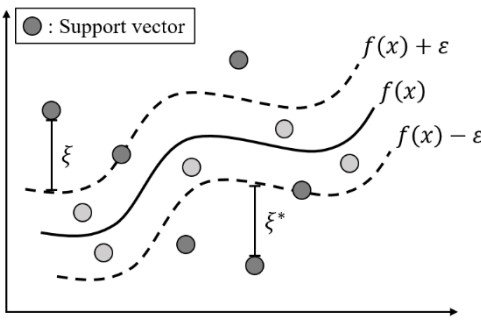

b) Random Forest (RF)

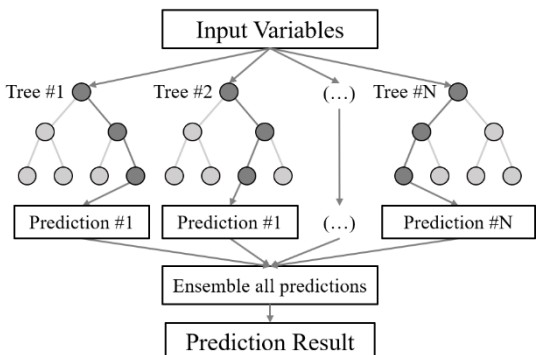

c) Gradient Boosting Machine (GBM)

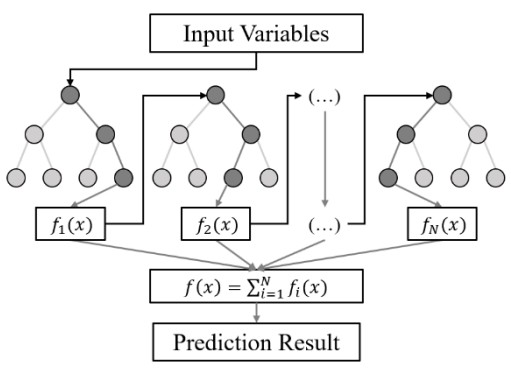

d) $k$-Nearest Neighbor (kNN)

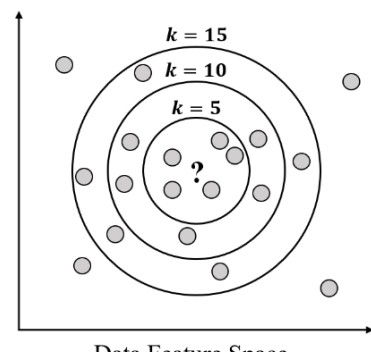

e) Artificial Neural Network (ANN)

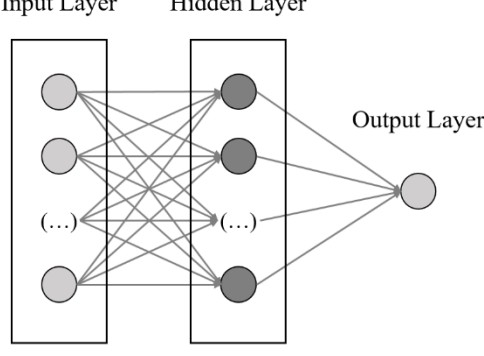

f) Multiple Linear Regression (MLR)

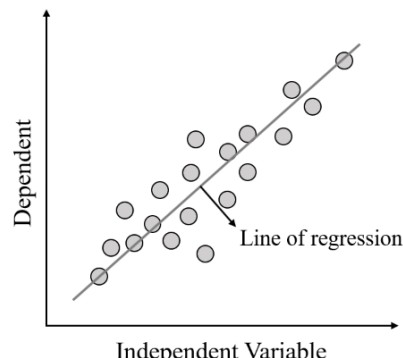

**Figure 3: Schematic of each machine learning method: SVR (a), RF (b), GBM (c), kNN (d), ANN (e), and MLR (f).**



## 4. Results

### 4.1 Training and validation results of machine learning methods

Figure 4 shows the cloud cover prediction results obtained using the training set for each machine learning method. The hyper-
parameters were optimized using the training and validation sets. The higher the frequency in the diagonal one-to-one boxes, the better the agreement between the observed and predicted cloud cover. In other words, the closer the diagonal one-to-one boxes are to purple or red, the higher is the agreement. For the training set, the highest human-eye observation data frequency by cloud cover was 26.80% at 0 tenth; this was followed by 19.97% at 10 tenths and 3.65–11.92% at 1–9 tenths. For the SVR model, the 0- and 10-tenths frequencies were 19.27% and 18.40%, respectively, being the greatest agreement among the machine learning models. As detailed in Table 3, the SVR accuracy, recall, and precision for all cloud cover were 0.94, 0.70, and 0.76, respectively, indicating the best prediction performance. The accuracy was in the range of 0.91–0.98 for each cloud cover, whereas recall and precision were in the ranges of 0.42–0.92 and 0.24–0.99, exhibiting low predictive power in the partly cloudy case. The bias was 0.07 tenth, the RMSE was 1.05 tenths, and R was 0.96. In the case of the RF model, the 0- and 10-tenths frequencies were 16.10% and 16.56%, respectively, being lower than those of the SVR model; however, the prediction for 1–9 tenths exhibited high agreement to within ±1 tenth. The accuracy, recall, and precision were 0.93, 0.67, and 0.76, respectively, lower than SVR model, but the bias and RMSE were the smallest at 0.02 and 0.71 tenth, respectively, and the R value was the highest at 0.99. However, for the validation set, the SVR model prediction performance (accuracy: 0.88, recall: 0.41, precision: 0.51, bias: 0.06 tenth, RMSE: 1.51 tenths, R: 0.93) was better than that of RF model. In other words, the RF model exhibited a tendency to overfit in this study. The accuracy of these results exceeds that of the classification machine learning method (0.6–0.85) presented by Dev et al. (2016) using day and night image data, and are higher than or similar to the accuracy (0.91–0.94) achieved using the regression and deep learning machine learning methods proposed by Shi et al. (2019, 2021) for day and night image data. Apart from the SVR and RF methods, the machine learning methods exhibited similar frequency distributions; however, the accuracy, recall, precision, and R were lower and the RMSE were higher in the order of GBM, kNN, ANN, and MLR. In particular, the MLR model had very poor predictive power (accuracy: 0.75, recall: 0.08, precision: 0.78) for 0 tenth using the training set.



a) Support Vector Regression (SVR)

b) Random Forest (RF)

c) Gradient Boosting Machine (GBM)

d) *k*-Nearest Neighbor (kNN)

e) Artificial Neural Network (ANN)

f) Multiple Linear Regression (MLR)

**Figure 4: Scatter plots of observed cloud cover and that predicted by machine learning methods (SVR (a), RF (b), GBM (c), kNN (d), ANN (e), and MLR (f)) on the training set.**





**Table 3: Prediction performance for all cloud cover of machine learning methods using training and validation sets.**

| Model | Set | Accuracy | Recall | Precision | Bias | RMSE | R |
|-------|-----|----------|--------|-----------|------|------|---|
| SVR | Training | 0.94 | 0.70 | 0.76 | 0.07 | 1.05 | 0.96 |
|  | Validation | 0.88 | 0.41 | 0.51 | 0.06 | 1.51 | 0.93 |
| RF | Training | 0.93 | 0.67 | 0.76 | 0.02 | 0.71 | 0.99 |
|  | Validation | 0.86 | 0.35 | 0.53 | −0.03 | 1.55 | 0.92 |
| GBM | Training | 0.89 | 0.47 | 0.59 | −0.00 | 1.03 | 0.97 |
|  | Validation | 0.86 | 0.36 | 0.50 | −0.06 | 1.58 | 0.92 |
| kNN | Training | 0.88 | 0.41 | 0.57 | 0.15 | 1.41 | 0.94 |
|  | Validation | 0.87 | 0.37 | 0.51 | 0.12 | 1.78 | 0.90 |
| ANN | Training | 0.86 | 0.33 | 0.49 | 0.03 | 1.69 | 0.91 |
|  | Validation | 0.85 | 0.31 | 0.46 | 0.01 | 1.92 | 0.88 |
| MLR | Training | 0.84 | 0.27 | 0.46 | −0.02 | 1.90 | 0.88 |
|  | Validation | 0.84 | 0.27 | 0.46 | −0.02 | 1.94 | 0.87 |

The relative importance of the input variable of the SVR method, which exhibited the best predictive performance in this study, is shown in Fig. 5. $BRD_{Q4}$ had the highest relative importance at 8.54% whereas $RBR_{mode}$ had the lowest importance at 0.55%. 295  Among the RBR data features, $RBR_{Q0}$ had the highest importance at 7.06% and, among the Y data features, $Y_{Q0}$ had the highest importance at 3.78%. In terms of the cumulative relative importance, the BRD-, RBR- and Y-related data features contributed 38.25%, 31.44%, and 26.20% of the total (100%), respectively, to the cloud cover prediction, and the remaining data features contributed 4.10%.





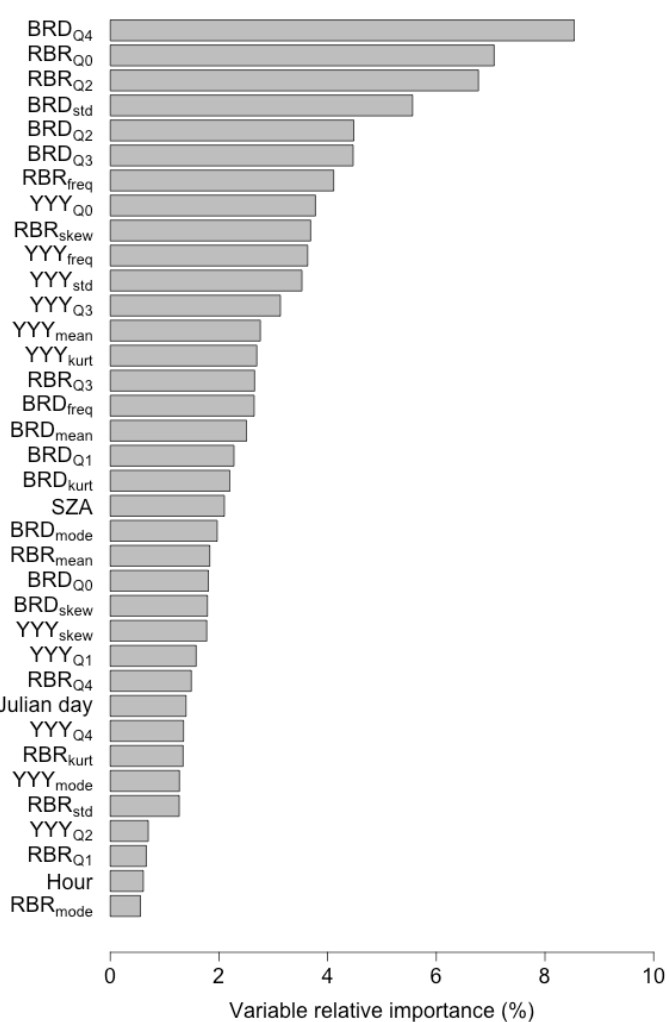


**Figure 5: Variable relative importance of SVR model on training set.**

## 4.2 Test set results for SVR model

Figure 6 shows the total cases and seasonal scatter plots of the DROM cloud cover and the ACOS cloud cover prediction

calculated from the SVR model using the test set. In the Korean Peninsula, the winter cloud cover is sparse (<5 tenths) as the weather is generally clear because of the Siberian air mass. In summer, the rainy season is concentrated under the influence of the Yangtze-River and Pacific air masses, and the cloud cover is dense (>5 tenths) until fall because of typhoons (Kim et al., 2018a, 2020a,). Furthermore, the Korean Peninsula experiences a westerly wind, cumulus heat generated in the western sea moves inland, and the cloud cover changes rapidly and continuously (Kim et al., 2021). The cloud cover distributions



calculated for all test set cases exhibited good agreement with the observed cloud cover, with accuracy, recall, and precision
of 0.88, 0.42, 0.52, respectively. Further, the bias, RMSE, and R were 0.04 tenth, 1.45 tenths, and 0.93, respectively. In fall,
the bias, RMSE, and R were –0.12 tenth and 1.30 tenths, and 0.95, respectively, indicating that the difference between the
observed and calculated cloud cover was small. In winter and summer, the RMSE was larger and R was lower than in the other
seasons. This is because the cloud cover calculation error is large at sunrise and sunset ($100° \geq SZA > 80°$), i.e., where daytime

($SZA \leq 80°$) and nighttime ($SZA > 100°$) intersect (Lalonde et al., 2010; Alonso et al., 2014; Kim et al., 2020b).
For the test set daytime cases, the bias, RMSE, and R were 0.10 tenth, 1.20 tenths, and 0.95, respectively, and 0.08 tenth, 1.59
tenths, and 0.93, respectively, for the night data. However, for sunrise and sunset, these values were –0.22 tenth, 1.71 tenths,
and 0.90, respectively. Relatively, the bias and RMSE were large and R was low. In spring and autumn, sunrise and sunset
images were learned at similar times (sunrise: 0600–0700 LST, sunset: 1800–1900 LST); however, differences between the

winter (sunrise: 0700–0800 LST, sunset: 1700–1800 LST) and summer (sunrise: 0500–0600 LST, sunset: 1900–2000 LST)
results are apparent because sunrise and sunset occurred late or early and exhibited different features from the data features
learned for those times (Liu et al., 2015; Li et al., 2019). That is, owing to the sunrise/sunset glow, high cloud cover calculation
errors are obtained at sunrise/sunset, when it is difficult to distinguish between the sky and clouds because of the reddish sky
on a clear day and the bluish cloud on a cloudy day (Kim et al., 2021). Therefore, for the test set, the bias, RMSE, and R for

sunrise and sunset in spring and autumn were –0.24 tenth, 1.46 tenths, and 0.93, respectively. However, in winter and summer,
the bias, RMSE, and R were –0.21 tenth, 1.93 tenths, and 0.86, respectively. Nevertheless, the results of this study surpass
those of Kim et al. (2016) for daytime (0800–1700 LST; bias: –0.36 tenth, RMSE: 2.12 tenths, R: 0.87) and Kim et al. (2020b)
for nighttime (1900–0600 LST; bias: –0.28 tenth, RMSE: 1.78 tenths, R: 0.91) cases. Shields et al. (2019) employed different
day and night cloud cover calculation algorithms. In that approach, cloud cover calculation errors may occur at sunrise and

sunset. Therefore, if a day and night continuous cloud cover calculation algorithm is considered, the calculation error for this
discontinuous time period should be reduced (Huo and Lu, 2009; Li et al., 2019). Figure 7 shows the daily mean cloud cover
results based on the observed and calculated cloud cover for the test set. For the observed and calculated cloud cover, a bias
of 0.03 tenth, RMSE of 0.92 tenth, and R of 0.96 were obtained. The coefficient of determination ($R^2$) was 0.92, and the result
calculated from the SVR model constructed in this study explained approximately 92% of the observed data in the test set.






a) Total cases

b) Winter                                    c) Spring

d) Summer                                    e) Fall

**Figure 6: Scatter plots of total (a) and seasonal (b–e) cloud cover based on observed (DROM) and calculated (ACOS) cloud cover for the test set.**





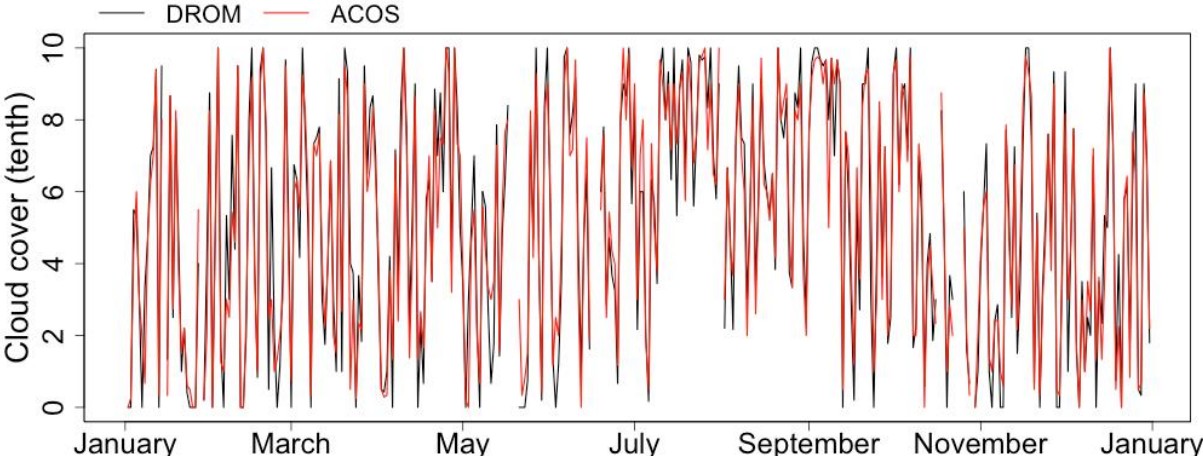

**340**    **Figure 7: Daily mean time series of observed (DROM) and calculated (ACOS) cloud cover for the test set.**

Figure 8 shows the frequency distribution of the differences between ACOS and DROM by season and time. In this frequency distribution, the higher the 0 tenth frequency, the higher the agreement between the observed and calculated cloud cover. The highest 0 tenth frequency was obtained in winter (46.05%) and the lowest in spring (35.58%), but 41.69% agreement was

**345**    obtained for all seasons. By time, high agreement of approximately 44% was obtained for both daytime and nighttime, but the lowest agreement (30.49%) was obtained for sunrise and sunset. In general, a difference of approximately 2 tenths from the observed cloud cover was obtained for the cloud cover calculated based on the ground-based imager data (Kazantzidis et al., 2012; Kim et al., 2016, 2020b; Wang et al., 2021). When a difference of up to 2 tenths was allowed between the observed and calculated cloud cover, the agreement was 90.95%, as detailed in Table 4. When a difference up to 1 tenth between both cloud

**350**    cover results was allowed for all cases, the agreement was 79.05%. When the difference was within 2 tenths, high agreement of 86.59% to 94.41% by season and by time was obtained. These results reveal greater agreement than those obtained by Cazorla et al. (2008), Kreuter et al. (2009), Kazantzidis et al. (2012), Krinitskiy and Sinitsyn (2016), Fa et al. (2019), Kim et al. (2016, 2020b), Xie et al. (2020), and Wang et al. (2021). In those works, 80–94% agreement was achieved when the allowed difference between the observed and calculated cloud cover was 2 oktas (2.5 tenths) or 2 tenths for day, night, and day and

**355**    night cases.



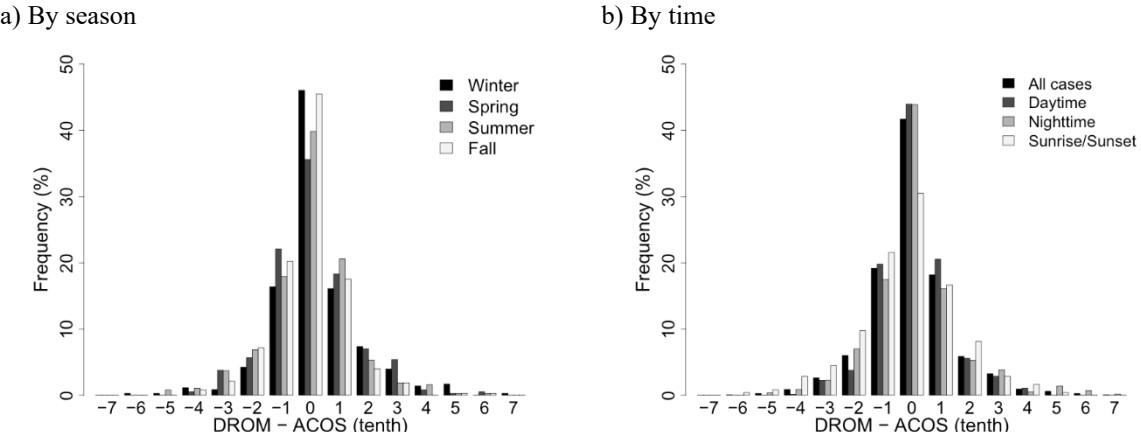

**Figure 8: Relative frequency distributions of differences between observed (DROM) and calculated (ACOS) cloud cover by season and time for the test set.**

**Table 4: Concordance frequency (%) according to the difference (Diff.) between the observed (DROM) and calculated (ACOS) cloud cover for the test set.**

| Diff. | Winter | Spring | Summer | Fall | Annual | Daytime | Night-time | Sunrise/Sunset |
|---|---|---|---|---|---|---|---|---|
| ±0 tenth | 46.05 | 35.58 | 39.84 | 45.48 | 41.69 | 43.96 | 43.88 | 30.49 |
| ±1 tenth | 78.53 | 76.01 | 78.36 | 83.24 | 79.05 | 84.29 | 77.45 | 68.70 |
| ±2 tenths | 90.11 | 88.68 | 90.50 | 94.41 | 90.95 | 93.66 | 89.69 | 86.59 |

## 5. Conclusions

In this study, data features of images captured using ACOS, a camera-based imager on the ground, were used in conjunction with machine learning methods to continuously calculate cloud cover for 24-h, at day and night. The data features of the images used as the machine learning input data were the mean, mode, frequency of mode, skewness, kurtosis, and quantile (Q0–Q4) of the RBR, BRD, and Y frequency distributions, respectively, along with the Julian day, hour, and SZA. The RBR, BRD, and Y data features were calculated through pre-processing using the methods described by Kim et al. (2020b) (masking removal

and distortion correction). These features indicate the sky and cloud colors depending on the light scattering characteristics in





the day and night, along with the presence or absence of clouds and the position of the sun (Heinle et al., 2010; Blazek and Pata, 2015; Li et al., 2019). The collected image data (100%) were composed of training (50%), validation (30%), and test (20%) sets, and were used for optimization of the models produced by the machine learning methods, comparative analysis of the prediction results of each machine learning method, and verification of the predicted cloud cover. In this study, the SVR,

RF, GBM, kNN, ANN, and MLR supervised machine learning methods were used. Among these methods, the SVR model exhibited the best prediction performance, with accuracy, recall, and precision of 0.94, 0.70, and 0.76, respectively. The cloud cover calculation results produced by the SVR on the test set had a bias of 0.04 tenth, RMSE of 1.45 tenths, and R of 0.93. With respect to this calculation result, when a difference of 2 tenths from the observed cloud cover was allowed, the agreement was 41.69%, 79.05%, and 90.95% for 0, 1, and 2 tenths difference, respectively.

Using the image data features and machine learning methods (best: SVM, worst: MLR) considered in this study, high accuracy cloud cover calculation can be expected; further, this approach is suitable for nowcasting. Based on the cloud information obtained from such cloud detection and cloud cover calculation, cloud physical properties such as the type, base height, optical thickness, and motion vector (Wang et al., 2016; Ye et al., 2017; Román et al., 2018; Zhang et al., 2018a) can be calculated. Ground-based observation of clouds using a camera-based imager, accompanied by cloud characteristic calculation, is an

economical method that can replace manned observations at synoptic observatories with automated (unmanned) observations. In addition, objective and low-uncertainty cloud observation is expected to be possible through widespread distribution of instruments such as those used in this study, to unmanned as well as manned observatories. Therefore, active research and development of imager-based cloud observation instruments is merited.

**Code availability:** The code for this paper is available from the corresponding author.

**Sample availability:** The sample for this paper are available from the corresponding author.

**Author contribution:** BYK carried out this study and the analysis. The results were discussed with JWC and KHC. BYK developed the machine learning model code and performed the simulations and visualizations. The manuscript was mainly written by BYK with contributions by JWC and KHC.

**Competing interests:** The authors declare that they have no conflict of interest.

**Acknowledgements:** This work was funded by the Korea Meteorological Administration Research and Development Program "Development of Application Technology on Atmospheric Research Aircraft" under Grant (KMA2018-00222).

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
