# Peer review of "24-hour cloud cover calculation using ground-based imager with machine learning"

_Atmospheric Measurement Techniques, 2021_

## Author Response (AR1)

**Reviewer #1**

This manuscript describes the use of several Machine Learning techniques in order to estimate the cloud cover from a series of statistical characteristics derived from all sky images. This study hardly shows any new concept, as analysis of all sky images have been conducted for about 15 years so far, including a great number of studies that use machine learning techniques. However, I tend to be favorable to the publication of this manuscript, as the explanation is quite clear, the number of analyzed ML techniques is high, the data set used is comprehensive (including day and night images), the way of reducing the information contained in an image is stimulating, and the results are relevant enough.

In any case, I have some few minor comments and suggestions that may be helpful to improve the paper.

Thank you for your consideration for reviewing this manuscript. Following the reviewer's comments, we have added the contents to the revised manuscript as detailed below. We believe that the quality of the manuscript has been improved and clarified through the reviewer's comments.

Line 25: I wouldn't say "to date", as it's been a while since several automated systems have been introduced, maybe not for standard (official) observations, but for uses of research or of solar plants management.

We have revised this sentence as follows:

Line 25: "In countries, including South Korea, that have not introduced automated systems, ground-based cloud cover observation has been performed using the human eye, in accordance with the normalized synoptic observation rule of the World Meteorological Organization (WMO), and recorded in tenths or oktas (Kim et al., 2016; Yun and Whang, 2018)."

Line 29. I would say that the use of "longer-period" observations is ambiguous here. I understand that you mean at a lower temporal resolution.

We have revised this sentence as follows:

Line 29: "Further, although continuous cloud cover observation during day and night is important, there is a lack of data continuity (observations with at least 1-hour intervals) because a person must perform direct observations (Kim et al., 2020b)."

Lines 42-42: I think that these two references regard only to camera systems; they could fit better in other places of the introduction (line 50, or line 57). Here there are other more general papers that I think are more adequate: Boers, R., M. J. De Haij, W. M. F. Wauben, H. K. Baltink, L. H. Van Ulft, M. Savenije, and C. N. Long, 2010: Optimized fractional cloudiness determination from five ground-based remote sensing techniques. J. Geophys. Res. Atmos., 115, 1–16, doi:10.1029/2010JD014661.

Lines 43 and 58: We have moved and added these references.

Line 46: regarding the use of ceilometers for cloud characterizations, this reference is also adequate here: Costa-Surós, M., J. Calbó, J. a. González, and C. N. Long, 2014: Comparing the cloud vertical structure derived from several methods based on radiosonde profiles and ground-based remote sensing measurements. Atmos. Meas. Tech., 7, 2757–2773, doi:10.5194/amt-7-2757-2014.

Line 47: We have added this reference.

Line 58: I think that when you say "cloud cover" you mean "pixel", here.

Line 60: We have revised this word.

Line 64: regarding the difficulties for distinguishing clouds from clear sky depending on some conditions, this reference may be useful: Calbó, J., C. N. Long, J. González, J. Augustine, and A. Mccomiskey, 2017: The thin border between cloud and aerosol : Sensitivity of several ground based observation techniques. Atmos. Res., 196, 248–260, doi:10.1016/j.atmosres.2017.06.010.

Line 64: We have added this reference.

Lines 76-77: there is some text which is repeated from line 74 in the same sentence.

We have revised this sentence as follows:

Line 75: "In this study, cloud cover was calculated continuously for 24-h from image data obtained by a camera-based imager on the ground, using image data features and machine learning methods. ANN, GBM, *k*-nearest neighbor (kNN), multiple linear regression (MLR), support vector regression (SVR), and random forest (RF) methods suitable for nowcasting were used for the calculation."

Line 119. I understand that this number of 7,402 images refers to images and concurrent human observations.

We have revised this sentence as follows:

Line 119: "Considering this, a total of 7,402 images of concurrent human observations were collected, excluding missing cases, from the ACOS."

Line 147-148: this method of masking (brightness greater than 240) may be valid for bright objects (Sun) but not for trees or building, if I understand correctly.

We have revised this sentence as follows:

Line 151: "Second, surrounding masks such as buildings, trees, and equipment, as well as light sources such as the sun, moon, and stars, were removed from the image (building, tree, and equipment: masking was performed when the mean RGB brightness was less than 60 in the daytime on a clear day; light source: masking was performed when the mean RGB brightness exceeded 240)."

Lines 151-152. I would say that the calculation of Y should be presented in section 2.2, not here.

Line 113: We have moved this sentence.

Line 155. I suggest stressing the fact that only the statistical characteristics (along with time information) are used as inputs for the ML methods.

We have revised this sentence as follows:

Line 158: "Statistical characteristics of the mean, mode, frequency of mode, kurtosis, skewness, and quantile (Q0–Q4: 0%, 25%, 50%, 75%, and 100%) data obtained for each frequency distribution were used as input for each machine learning method. As input data for machine learning, time information (Julian day and hour) allowed differentiating seasons and day and night. Further, SZA should be considered because the colors of the sky and clouds change according to the position of the sun (Blazek and Pata, 2015; Cazorla et al., 2015; Azhar et al., 2021)."

Eq. (7-11). I would write the equations much closer to the place where they are mentioned in the text (in line 146) not here after line 160.

Line 146: We have moved these equations after line 146.

Lines 168-169. You should specify that these values of 0, 10, and 5 tenths refer to the sample images in Fig. 2.

We have revised this sentence as follows:

Line 168: "Human-eye observations at DROM yielded cloud-free (Fig. 2a and 2g), overcast (Fig. 2b and 2h), and partly cloudy (Fig. 2c and 2i) case values of 0, 10, and 5 tenths, respectively."

Section 3. This section is difficult to judge by me, as I'm an atmospheric scientist, not a computational science researcher. However, I would suggest presenting the ML methods in increasing complexity order. That is MLR as the first method, then K-nearest, SVR, etc.

Line 200: We have arranged this section and figure 3.

Fig 4 and fig 6: I think that the color code is not clear. Of course, cloud cover classes which are more populated get higher percentages (i.e., colors tending to orange and red). But what is relevant is, for each cloud cover class (given by the reference, DROM), how are the ACOS predictions distributed. In other words, I would color the boxes regarding the percentage relative to each class, not to the overall number of cases in the whole dataset.

We have redrawn these figures and revised several sentences as follows:

Line 265: "Figure 4 shows the cloud cover prediction results obtained using the training set for each machine learning method. The hyper-parameters were optimized using the training and validation sets. Each box in the figure denotes the ratio (%) of the number of observations for each cloud cover in the DROM and the number of predictions for each cloud cover in the SVR model. The higher the frequency in the diagonal one-to-one boxes, the better the agreement between the observed and predicted cloud cover. In other words, the closer the diagonal one-to-one boxes are to red (i.e., 100%), the higher is the agreement."

Line 271: "For the SVR model, the 0- and 10-tenths frequencies were 71.88% and 92.15%, respectively, being the greatest agreement among the machine learning models."

Line 276: "In the case of the RF model, the 0- and 10-tenths frequencies were 61.79% and 80.65%, respectively, being lower than those of the SVR model; however, the prediction for 1–9 tenths exhibited high agreement to within ±1 tenth."

Line 291: "The number of observations for each observed cloud cover are 0: 992, 1: 136, 2: 141, 3: 156, 4: 149, 5: 135, 6: 221, 7: 271, 8: 320, 9: 441, and 10: 739."

Line 349: "Parentheses values are the number of observations for each cloud cover in DROM."

Fig 5. I find this figure quite interesting, as it allows some physical interpretation of the results shown. So, despite some comments are made in lines 293-298, I would suggest adding more detailed comments if possible. Do you think that RBR ad RBD are "redundant" information? Why date and hour have a minor role?...

We have added the explains as follows:

Line 301: "The relationship between input data features is complex to determine the optimal hyperplane of the SVR model, and the variable importance is determined so that the cloud cover can be calculated with the smallest

error using the observed cloud cover (Singh et al., 2020). Even if the BRD-related data features have the same RBR characteristics, they contribute to machine learning more comprehensively by day, night, and cloud presence depending on the BRD value; therefore, they are critical to cloud cover calculation. By contrast, the Y-related data feature is sensitive to the RGB brightness (especially the G brightness) in the image, but the difference in the Y characteristics according to the cloud cover during the day was not large; thus, their importance was relatively low. Although time information and SZA can provide information such as daytime, nighttime, and sunset/sunrise images, they have the lowest importance because they do not have statistical characteristics that can be used to directly calculate cloud cover. The importance of these data features may vary depending on the camera's sensor (Kazantzidis et al., 2012)."

Line 346. When you say "in general", do you mean "in other, previous studies"?

We have revised this sentence as follows:

Line 358: "Previous studies obtained a difference of approximately 2 tenths from the observed cloud cover for the cloud cover calculated based on the ground-based imager data (Kazantzidis et al., 2012; Kim et al., 2016, 2020b; Wang et al., 2021)."

Line 380-384: I don't see how come you can get physical characteristics of clouds, as this study only predicts clouds cover, but no spatial characteristics. You could extend a little this suggestion for further work.

We have revised this sentence as follows:

Line 392: "Based on the cloud information obtained from such cloud detection and cloud cover calculation makes it possible to calculate the physical properties of various clouds (Wang et al., 2016; Ye et al., 2017; Román et al., 2018; Zhang et al., 2018a). In other words, it is possible to calculate cloud base height and cloud motion vector through the geometric and kinematic analysis of continuous images using single or multiple cameras (Nguyen and Kleissl, 2014), which can be used for cloud type classification according to cloud cover, cloud base height, and cloud color feature (Heinle et al., 2010; Ghonima et al., 2012)."

**Reviewer #2**

This manuscript presents a comparison of several methods of obtaining the cloud cover without reliance on human eye. The presentation of the methods and the results is well-written, clear and convincing. As far as I can tell, it does not introduce any new tools to obtain cloud cover via machine learning or other methods, but it does provide a valuable benchmark using a large sample of images.

Thank you for your consideration for reviewing this manuscript. Following the reviewer's comments, we have added/revised the contents/sentences to the revised manuscript as detailed below. We believe that the quality of the manuscript has been improved and clarified through the reviewer's comments.

I only have a few minor suggestions:

Line 164: How is image radius defined ? Why aren't the equations placed after line 146 when they are first mentioned ?

We have added definition as follows:

Line 148: "…, *radi* is the image radius (distance between center and edge pixel of circular images), …"

Line 146: And we have moved these equations after line 146.

Line 207 : Missing "was" :    ... was used, the SVR kernel ...

Line 228: We have revised this sentence.

Line 212 : -> where x_i is the i-th data point ...

We have revised this sentence as follows:

Line 232: "where subscript *i* and *j* are i-th and j-th data points, respectively, and …"

Line 345 : By time -> Conditioned on the time of day, high ...

Line 357: We have revised this sentence.